# Cerebral Apolipoprotein D Exits the Brain and Accumulates in Peripheral Tissues

**DOI:** 10.3390/ijms22084118

**Published:** 2021-04-16

**Authors:** Frederik Desmarais, Vincent Hervé, Karl F. Bergeron, Gaétan Ravaut, Morgane Perrotte, Guillaume Fyfe-Desmarais, Eric Rassart, Charles Ramassamy, Catherine Mounier

**Affiliations:** 1Laboratoire du Métabolisme Moléculaire des Lipides, Centre de Recherches CERMO-FC, Département des Sciences Biologiques, Université du Québec à Montréal (UQAM), 141 av. du Président-Kennedy, Montréal, QC H2X 1Y4, Canada; desmarais.frederik@courrier.uqam.ca (F.D.); bergeron.karl-frederik@uqam.ca (K.F.B.); ravaut.gaetan@uqam.ca (G.R.); desmarais-fyfe.guillaume@courrier.uqam.ca (G.F.-D.); 2Laboratoire de Biologie Moléculaire, Département des Sciences Biologiques, Université du Québec à Montréal (UQAM), 141 av. du Président-Kennedy, Montréal, QC H2X 1Y4, Canada; Vincent.herve@inrs.ca (V.H.); rassart.eric@uqam.ca (E.R.); 3Centre Armand-Frappier Santé Biotechnologie, Institut National de la Recherche Scientifique (INRS), 531 boul. des Prairies, Laval, QC H7V 1B7, Canada; morgane.perrotte@iaf.inrs.ca

**Keywords:** apolipoprotein D, basigin, protein accumulation, blood-brain barrier

## Abstract

Apolipoprotein D (ApoD) is a secreted lipocalin associated with neuroprotection and lipid metabolism. In rodent, the bulk of its expression occurs in the central nervous system. Despite this, ApoD has profound effects in peripheral tissues, indicating that neural ApoD may reach peripheral organs. We endeavor to determine if cerebral ApoD can reach the circulation and accumulate in peripheral tissues. Three hours was necessary for over 40% of all the radiolabeled human ApoD (hApoD), injected bilaterally, to exit the central nervous system (CNS). Once in circulation, hApoD accumulates mostly in the kidneys/urine, liver, and muscles. Accumulation specificity of hApoD in these tissues was strongly correlated with the expression of lowly glycosylated basigin (BSG, CD147). hApoD was observed to pass through bEnd.3 blood brain barrier endothelial cells monolayers. However, cyclophilin A did not impact hApoD internalization rates in bEnd.3, indicating that ApoD exit from the brain is either independent of BSG or relies on additional cell types. Overall, our data showed that ApoD can quickly and efficiently exit the CNS and reach the liver and kidneys/urine, organs linked to the recycling and excretion of lipids and toxins. This indicated that cerebral overexpression during neurodegenerative episodes may serve to evacuate neurotoxic ApoD ligands from the CNS.

## 1. Introduction

Apolipoprotein D (ApoD) is a 25–30 kDa glycosylated protein belonging to the lipocalin superfamily of hydrophobic molecule carriers [1,2,3,4]. ApoD is known for its ability to bind various ligands, including arachidonic acid (ARA) and progesterone [1,5,6]. ApoD can bind ARA and mediate its release from cell membranes. Furthermore, since ARA is the precursor of the eicosanoid class of lipid inflammation mediators, ApoD’s capacity to bind ARA allows it to modulate the production of eicosanoids and attenuate inflammation [7,8]. ApoD can also limit inflammation and oxidative stress by reducing oxidized lipids [1,9,10,11,12]. Because it is overexpressed during many neurodegenerative diseases and stresses, ApoD is considered an important factor in brain protection and repair [9,11,13,14,15]. ApoD is also a member of the apolipoprotein family as it associates with lipoproteins (mainly high-density lipoproteins) in the blood. In humans, contrary to most apolipoproteins, ApoD is minimally produced in the liver and intestines [1,5]. Rather, mRNA expression is found, although in varying quantities, in several tissues including the central nervous system (CNS), mammary glands, spleen, adipose tissues, adrenals, and skin [1,16,17]. In contrast to humans, in mice and rats, ApoD mRNA expression is mainly restricted to the CNS [1,2,18,19,20]. However, moderate to high protein levels are found in peripheral tissues including the liver, suggesting that the protein could cross the blood brain barrier (BBB) [1,21].

In support to the hypothesis that apoD can exit the CNS, the overexpression of human ApoD (hApoD) in the CNS of transgenic mice leads to an elevated level of apoD in the livers, likely originating from expression in the CNS [19]. These transgenic mice develop hepatic and muscular steatosis [8,19,22] probably because apoD modulates the metabolism of these tissues.

Beside its ability to circulate as part of lipoproteins, up to 10% of plasma ApoD is found in the protein phase, indicating that ApoD may also circulate freely in its soluble form [23]. Three recent studies also highlighted the presence of ApoD in extracellular vesicles [24,25,26]. ApoD has been shown to be transported to neurons by extracellular vesicles originating from astrocytes [26]. ApoD can also be found associated with cerebrospinal fluid lipoproteins [27,28,29]. ApoD was also observed in urine [30]. Together, the results show that ApoD transport is polyvalent and complex.

The Basigin (BSG) receptor, also known as cluster of differentiation 147 (CD147), mediates ApoD internalization in neurons [14]. BSG is also responsible for the internalization of other proteins such as caveolin with its affinity being dependent on BSG glycosylation level [31]. BSG is expressed in a wide range of tissues and cell types [32] (data available at https://www.proteinatlas.org/ENSG00000172270-BSG/tissue) (accessed on 25 November 2020). Circulating ApoD, through BSG-mediated internalization, could therefore accumulate in many different structures. BSG is expressed in mouse brain endothelial cells, the main component of the blood-brain barrier (BBB) [33]. The BBB is the largest barrier separating the peripheral circulation from the CNS [34]. It is a highly selective barrier, composed of tightly linked endothelial cells that cover the bloodstream capillaries. Astrocytes and pericytes also participate in the architecture and function of the BBB. In addition to BSG, apolipoprotein receptors such as Low-density lipoprotein receptor (LDLR), LDLR-related protein (LRP1) and Megalin are also expressed in BBB endothelial cells and are involved in transporting ligands across the barrier [35].

The mouse brain microvascular endothelial cell line bEnd.3 is the most characterized in terms of tight junction proteins [36,37] and is thus often used to study the endothelial component of the BBB [38]. These cells express the tight junction proteins ZO-1, occludin, and claudin-5 [39] and possess fluorescein permeability similar to those of primary mouse endothelial cells [40]. These mouse brain capillary endothelial cells contain several receptors present on the surface of endogenous BBB endothelial cells, including apolipoprotein receptors such as LDLR and LRP1. These cells also possess receptors that participate in the transcytosis process of proteins such as transferrin and lactoferrin [35].

In this study, we endeavor to determine if cerebral ApoD can reach the bloodstream from the brain and if BSG plays a role in this process. We also set out to determine in which peripheral organs ApoD preferentially accumulates once in circulation.

## 2. Results

### 2.1. hApoD Exits the Central Nervous System and Reaches Peripheral Tissues

We endeavored to establish whether hApoD can exit the brain and reach the bloodstream and accumulate in peripheral organs. For this, the cerebral ventricles of mice were injected with radiolabeled hApoD or albumin as a control. Injected proteins were allowed to diffuse from the ventricles and tissue/fluid for 3 or 6 h. Samples were subsequently taken throughout the mouse body. During this procedure, mice were perfused with a saline solution to avoid contamination in tissues stemming from potential radiolabeled proteins in the blood. Also, BBB integrity was confirmed by circulating an Evans blue solution in perfused mice (Appendix A). As expected, 3 h after ICV injection of radiolabeled proteins, most of the recovered albumin (93%; Figure 1A), remained sequestered in the brain. Within this time frame, 41% of the recovered hApoD was found in the periphery, showing that hApoD can exit the brain compartment. Interestingly, a significant proportion of hApoD accumulated in the liver (6.06%) but the major fraction was found in urine (28.87%). Radioactivity levels in the bloodstream were very low, indicating that hApoD was rapidly cleared from the circulation. Electrophoresis of urine and brain protein samples in denaturing conditions confirmed that the radiation detected corresponded to native hApoD molecules migrating at 30 kDa (Figure 1B).

### 2.2. hApoD Accumulates in Specific Tissues

ICV injections were limited to a total volume of 10 μL (5 μL per ventricle), adding up to approximately 2.5 μg of radiolabeled proteins. Under these experimental restrictions, hApoD could only be shown to accumulate in the kidneys and liver (Figure 1A). The radiation found in other organs (muscles, pancreas, spleen, intestines, and others) was too close to background radiation levels to confidently affirm that hApoD had accumulated in these organs. To determine if hApoD can target other tissues, we injected a greater amount of radiolabeled protein (approximately 10 μg) directly in the bloodstream. Note that this higher dose of hApoD was still within physiological parameters for circulating ApoD (Rassart et al. 2000). Albumin was again chosen as a control because of its propensity to remain in the circulation (Andersen et al. 2014). Again, the blood and its radioactive content were eliminated by perfusion with a saline solution before tissue samples were taken. The amount of recovered radiolabeled hApoD was evaluated 3 and 6 h post injection (Figure 2A) and as a function of the albumin control (Figure 2B). Accumulation of hApoD occurred at a higher rate than albumin in the kidneys (5.6-fold), liver (30.0-fold), pancreas (3.1-fold), caecum (2.8-fold), spleen (3.3-fold), mesenteric adipose tissue (MAT; 4.2-fold), and muscles (2.4-fold) at least at either 3 or 6 h. The kidneys emerged as the tissue that most readily accumulated hApoD with 14% of recovered radioactivity per gram of tissue. This accumulation was probably due to hApoD within the nephrons since a major part of the radioactivity was found in urine (Figure 3A). The biggest difference between albumin and hApoD accumulation; however, was found in the liver (Figure 2B), indicating a very strong specificity of hApoD for this tissue. Interestingly, the accumulation of hApoD in muscles was stronger than albumin 3 h after injection. However, after 6 h, hApoD levels diminished (from 1.6 to 1.0% per gram of tissue), while albumin levels increased (from 0.7 to 1.4% per gram of tissue). Conversely, hApoD accumulation in the MAT appeared to be more persistent than albumin. MAT radiation from hApoD only diminished by 0.10% per gram of tissue in 3 h. In comparison, radiation from albumin was reduced by 0.96% per gram of tissue in that same interval. Very interestingly, hApoD within the bloodstream did not appear to enter the brain, suggesting that hApoD can only cross the BBB in one direction.

To establish a general portrait of hApoD biodistribution when it exits the bloodstream, we extrapolated the total radioactivity found in each organ (or fluid) according to their total weight (or volume). Since we did not measure the total weight of adipose tissues nor of bone marrow, these tissues were excluded from this analysis. Albumin was mainly found in circulation at both 3 (91%) and 6 h (83%) after injection (Figure 3A). As expected from our previous results, hApoD exited the bloodstream at a higher rate than albumin. After 6 h, half of the injected hApoD had left the circulation and a significant fraction was again found in urine (33%; Figure 3A). Globally, hApoD accumulation level in tissues was higher at 3 h (25.2%) compared to 6 h (18.1%), suggesting a progressive shift of hApoD from blood to tissues and finally to urine. The principal target of hApoD at 3 h (Figure 3B) was the muscles (10.67%) followed by the liver (8.93%), kidneys (3.09%), pancreas (1.24%), and spleen (0.41%). Because of their mass, muscles accounted for a large proportion of hApoD accumulation despite their poor apparent affinity for hApoD (Figure 2). For instance, while the liver accumulated hApoD at a much higher rate (6.0% of recovered radioactivity per gram of tissue), it was still outclassed in total hApoD accumulation by the muscles despite their low accumulation rate (1.6% of recovered radioactivity per gram of tissue; see Figure 2A).

### 2.3. hApoD Transcytoses through bEnd.3 Brain Endothelial Barrier Cells

Since hApoD can exit the brain compartment, we explored the possibility that hApoD might be able to pass through a BBB model consisting of a monolayer of the endothelial bEnd.3 cells. The impermeability of this monolayer model to passive diffusion was validated by electrical resistance measurement (≥35 Ohms/cm^2^) and Dextran permeability assay (Appendix A). Expression of Claudin-5 between adjacent cells was also validated, confirming the formation of tight junctions (Figure 4A). Within 24 h, exogenous hApoD applied to the top compartment of the monolayer was internalized (Figure 4A) and was found in the bottom medium (Figure 4B), indicating that hApoD is subject to transcytosis through bEnd.3 cells in a dose-dependent manner. However, bEnd.3 cells are not polarized [41,42]. Therefore, our results may have underestimated the amount of hApoD that actually underwent transcytosis since part of the internalized protein could be released back to the top compartment where it was first internalized and not contribute to the buildup of ApoD concentration in the bottom compartment.

### 2.4. Cyclophilin A Competition Does Not Reduce hApoD Internalization in Brain Endothelial Barrier Cells

Having confirmed that hApoD can cross through bEnd.3 brain endothelial cells, we next endeavored to determine if the BSG receptor was implicated in this process. We used cyclophylin A, a BSG ligand known to compete with ApoD for BSG-dependent internalization [14]. We first confirmed that cyclophilin A was non-toxic at the concentrations used (Figure 5A). Surprisingly, internalization and transcytosis of hApoD by bEnd.3 monolayers were unchanged despite the presence of various molar ratios of competing cyclophilin A (up to 10-fold excess relative to hApoD; Figure 5B–D). We then determined if BSG was expressed in bEnd.3 cells in comparison to control tissues (brain and liver). Our analysis showed that BSG was expressed in bEnd.3 cells at a comparable level to the liver, the tissue with the highest specific affinity for hApoD. Interestingly, BSG expression was much stronger in bEnd.3 than in whole brain lysate. Additionally, the level of BSG glycosylation varied greatly between samples (Figure 5E). BSG glycosylation can influence its protein–protein interactions [31]. Interestingly, the ratio of lowly-glycosylated BSG (LG-BSG) relative to total BSG expression appeared to align with our observed cellular affinity for hApoD internalization in vivo (Figure 2A and Figure 5F). The liver, one of the tissues that most readily accumulates hApoD, had a high LG-BSG /total ratio, while the bEnd.3 cells had a lower ratio and needed 24 h to internalize and excrete hApoD to the bottom media. This suggested a greater role of BSG in peripheral tissues than in endothelial cells for hApoD internalization.

### 2.5. Relationship between BSG Glycosylation and hApoD Accumulation

To further explore if BSG glycosylation was linked to the degree of tissue hApoD internalization, we studied the expression of the multiple glycosylated forms of BSG in mice tissues. BSG expression, per mg of tissue, was heterogeneous between tissues, with the liver and heart presenting an especially high overall BSG expression and adipose tissues having a low expression (Figure 6A). BSG bands were observed at multiple size and in varying patterns according to the tissue type. The SAT, muscles, and the brain only expressed the highly-glycosylated form of BSG (HG-BSG, >48 kDa). The liver and pancreas, however, expressed multiple forms of BSG (Figure 6A). We next performed correlations between BSG expression by glycosylation levels and the specific accumulation of hApoD in each tissue. As expected, the total expression of BSG in tissues was positively correlated with hApoD accumulation. However, the correlation was only statistically significant for the 3-h time point (Figure 6B). Interestingly, expression of the HG-BSG form was not found to be associated with hApoD-specific accumulation. In opposition, LG-BSG expression was correlated with hApoD specific accumulation at both time points, but especially 6 h after injection (*p* = 0.0047 **). Kidneys were outliers relative to all other tissues investigated and were therefore excluded from these correlation analyses (presented as separate data set) (Figure 6B). It is very likely that hApoD does not require to be internalized in kidney cells during urine excretion and it is probably passively filtered through the glomerular pores (discussed later).

## 3. Discussion

Our results demonstrated that injected hApoD can exit the brain and reach peripheric tissues. These observations were made by comparing the amount of hApoD that escaped the CNS to an albumin control. The rate at which hApoD leaves the brain was observed to be higher than that of albumin. Albumin is a natural component of the cerebro-spinal fluid (CSF). Its concentration in the CSF is lower than 1/125th of the one found in circulation (CSF/serum ratio < 8 × 10^−3^) [43] and the CNS barriers greatly limit albumin’s ability to transfer bilaterally between the brain and the circulation. Similarly to other components of the CSF, albumin is constantly eliminated from the brain by the brain’s waste clearance system [43,44]. The elimination of albumin from the CNS is a passive process that does not require an active transport across CNS barrier. In contrast, the hApoD’s higher exit rate strongly suggested that it might be actively transported.

The two major circulating hApoD recipient organs are the kidneys/urine and the liver, the first being implicated in lipid metabolism and recycling and the latter being mostly implicated in waste removal. This corroborated our previous hypothesis that cerebral hApoD reaches the liver [8]. We previously showed that hApoD overexpression in the brain results in the development of an hepatic and muscular steatosis in mice [19]. The presence of hApoD as well as an increase in ARA proportion in the liver suggested that hApoD derived from the brain was responsible for the added efflux of ARA into the liver [8,19,22].

Surprisingly, a large part of the hApoD recovered outside the CNS was found in the urine. It was previously shown that ApoD can pass through the kidneys and accumulate in urine in monomeric and dimeric form [45]. Multiple factors including protein size and charge determine which molecules can pass through the kidneys by glomerular filtration. It is generally believed that proteins and polymers in the range of 30–50 kDa can pass through the glomerular pores. However, the diameter and shape of the molecules are also important as elongated molecules such as 350–500 kDa nanotubes with 1 nm diameter have been shown to be efficiently cleared by glomerular filtration [46]. Glomerular pores have been described as cuboids with an average dimension of 4 (40 Å) by 14 nm (140 Å) in a cross section, and 7 nm (70 Å) in length [47]. Albumin is a 66–69 kDa flexible, ellipsoid-shaped protein with a 3.8 nm diameter and a length of 15 nm. Despite its size, approximately 3.3 g of albumin is filtered daily in the human kidney [48]. Therefore, it was not surprising to find albumin in urine. ApoD is a 29–32 kDa, 4.5 nm (45 Å) by 4.0 nm (45 Å) protein [49]. Considering its size, it is likely that hApoD is also able to directly pass through the glomerular pores and end up in urine without the assistance of active transport. This idea was supported by the fact that kidneys were outliers in our correlation between BSG expression and hApoD accumulation in tissues (Figure 6B). This indicated that BSG was likely not implicated in this process or that another mechanism was far more important for hApoD accumulation in kidney\urine. However, the physiological relevance ApoD’s capacity to be filtered by the kidneys remain elusive.

One possible interpretation is that ApoD could also participate in the elimination of harmful waste products (oxidized lipids) and pro-inflammatory molecules (ARA) from the CNS or other targeted tissues. ApoD is overexpressed in periods of neurodegenerative stress [15,50,51,52,53,54,55]. ApoD is known to facilitate myelin clearance, extracellular matrix remodeling and axon regeneration after nerve injury [17]. A large quantity of hydrophobic and pro-inflammatory molecules is released from damaged neurons and myelin sheets after nerve injury. ARA and lysophospholipids (LPC) are two of the major products resulting from PLA2-mediated myelin cleavage and are both ApoD ligands. In fact, ApoD-null mice have higher basal levels of free ARA and LPC in intact nerves as well as a lower free LPC level than wildtype mice in injured nerves [17]. The lack of ApoD during nerve injury appears to cause a heightened inflammatory response, delayed inflammation resorption, and slower healing process [17]. It is believed that this occurs, at least in part, because of a lack of proper management and clearance of free ARA and LPC. Previous studies on ApoD’s role in managing these substances only looked at the possibility that these harmful compounds were sequestered by ApoD and reabsorbed locally by cells in the CNS. Considering our present results, however, it is possible that ApoD might redirect part of the released lipids away from the neurons and into the kidneys and liver during these episodes. This may prevent further damage caused by peroxidized lipids or pro-inflammatory ARA-derived eicosanoids in the brain [17,56].

Our results also showed that hApoD is rapidly depleted from the circulation in favor of tissular accumulation or excretion via the kidneys/urine. Interestingly, while ApoD is known to be increased in the CSF of patients with Alzheimer’s disease [51], its plasmatic levels do not appear to be affected [57]. This could be due to the quick pace at which hApoD is depleted from plasma. Unfortunately, no data exist on ApoD concentration in the urine and peripheral tissues from Alzheimer and other neurodegenerative diseases patients.

In addition to tissues with lipid recycling and excretory functions, a number of other tissues were targeted by circulating ApoD. The skeletal muscles were also an important reservoir of hApoD accumulation, despite their poor specific affinity for hApoD (Figure 3A) and low expression of the highly glycosylated BSG (Figure 6A). They only remained an important hApoD reservoir because of their large mass. Additionally, accumulation of hApoD over albumin was only significant 3 h post-injection and tended to be lower than albumin at 6 h post-injection. It is worth mentioning that, while albumin largely stays in circulation, it can also accumulate in muscles [58] and intestinal cells [59]. This could explain why hApoD accumulation was stronger than albumin in these tissues at only one time point. Another evidence that ApoD targets the skeletal muscles is that transgenic mice overexpressing hApoD in their CNS develop muscle steatosis [19]. The function of ApoD in the skeletal muscles is likely unrelated to ApoD’s hypothetical function of removing harmful molecules from the CNS. Other authors described ApoD as a lipid transporter with a very large diversity of ligands. In fact, ApoD can bind cholesterol and pregnenolone, bringing these essentials lipids in various organs where they can be used. Previous studies also mentioned that increased expression of ApoD in the CNS during brain injury can increase the availabilities of essential lipids for the synthesis of new cellular membranes (Reviewed in Rassart et al., 2020 [1]).

As of now, the literature provides little insight into ApoD’s potential role in skeletal muscles. One study reported that ApoD, followed by the leptin receptor (LEPR), are the two most upregulated transcripts in muscle disuse atrophy. Meanwhile, genes responsible for energy metabolism, mitochondrial function, cell cycle regulation, stress response, sarcomere structure, cell growth/death, and protein turnover were downregulated [60]. ApoD is also upregulated in age-related skeletal muscle cells senescence, where it is expected to play anti-oxidative and anti-inflammatory roles [1,9,61]. Our results indicate that circulating ApoD could also be implicated in these mechanisms.

The spleen also appears to be targeted by circulating hApoD. The spleen is the major site of heme recycling from senescent red blood cells. There, heme is converted to bilirubin and then transported to the liver to be secreted into bile in the intestines [62]. Interestingly, ApoD can bind to bilirubin [63] and has long been suspected to participate in heme recycling [5].

hApoD appeared to also have affinity for the MAT. We previously showed that ApoD protein levels in adipose tissue are linked to improved metabolic parameters in obese women. Specifically, protein levels of ApoD in the human MAT were associated with reduced circulating TNF-α and improved insulin sensitivity (QUICKY index). These correlations did not exist with ApoD mRNA expression [64] suggesting that circulating ApoD, after adipose tissue internalization, was specifically responsible for this association. This hypothesis is reinforced by our present results, which indicate that circulating hApoD can accumulate in the MAT. However, the role of exogenous ApoD in adipose tissues remains unclear. In this tissue, ApoD may exert anti-inflammatory effects through its ability to modulate ARA metabolism [7,8,22], to reduce oxidative stress [9,10,65,66,67], and to disrupt osteopontin function, a protein implicated in macrophage recruitment [1,68].

We used an in vitro model of the endothelial part of the BBB, bEnd.3 cell monolayers, to confirm that hApoD can be subject to internalization and transcytosis. Beside the BBB, there are also other barriers between the periphery and the CNS through which ApoD could cross. The blood-CSF barrier (BCSFB), for instance, separates the blood from the CSF. It is formed by epithelial cells and the tight junctions of the choroid plexus. ApoD could potentially cross this barrier, enter the bloodstream, and then be distributed to peripheral organs. However, the BBB area in the human brain is 10 times greater than the BCSFB area [69]. Furthermore, since ApoD is mainly secreted by glial cells [1], in proximity to the BBB, its preferred exit route is likely to be the BBB. This motivated the use of the bEnd.3 cell line as our in vitro model.

Considering that ApoD internalization is a BSG-dependent, cyclophilin A-sensitive mechanism in neurons and HEK 293T [14], we expected BSG to be implicated in hApoD internalization by bEnd.3 cells. Though BSG is expressed by bEnd.3 cells, cyclophilin A did not limit hApoD internalization/transcytosis in our hands. Interestingly, the BSG glycosylation pattern in bEnd.3 cells is different from the one reported in 293T cells. The 293T cells appear to possess a greater amount of LG-BSG (especially 30–35 kDa). Furthermore, hApoD internalization by 293T cells was reported as soon as 4 h after administration [14], while that process necessitated 24 h to be detectable in our bEnd.3 cell assays (data not shown). Our in vivo data showed that HG-BSG expression is not correlated to hApoD internalization in tissues, contrary to the LG-glycosylated forms. Higher levels of glycosylation present on a protein can hinder its protein–protein interactions [70]. In agreement with this observation, deglycosylation of BSG was reported to increase its interaction with Caveolin-1 [31]. It is likely that a similar phenomenon occurs between BSG and ApoD. This was supported by the fact that underglycosylated BSG expression is strongly associated with hApoD specific accumulation in peripheral tissues (Figure 6B). Taken together, these results suggested that, while BSG may be an important factor in peripheral hApoD cellular internalization, it may not be implicated in hApoD’s capacity to exit the brain.

These results may also imply that BSG is not the exclusive receptor for ApoD. Inhibiting LDLR has previously been shown to reverse the synaptogenic effects of ApoD in dorsal root ganglion cell cultures [71]. LDLR is present at the BBB as well as in bEnd.3 cells [35] and is a major ApoE receptor in the brain [72,73]. LDLR could therefore also participate in the passage of the ApoD through the BBB. The lack of competition by cyclophilin A could be due to an LDLR-mediated internalization of hApoD instead of a BSG-mediated one in bEnd.3 endothelial cells.

While we showed the passage of hApoD through endothelial cell monolayers, our model lacked the full complexity of the BBB. Though endothelial cells are the main components of the BBB, the mechanisms regulating the crossing of proteins through the BBB are more complex and involve other cell types. The BBB is also composed of pericytes and astrocytes in an architecture that is difficult to achieve in vitro. In fact, addition of glial cells to bEnd.3 cells in a co-culture model enhances the barrier function of the endothelial monolayer [38]. Of note, the presence of hApoD in the plasma was rapidly detectable in vivo (3 h), a kinetic that was not replicable in our in vitro model. It appears highly probable that other cell types are implicated in hApoD exit from the brain.

In our experiments, we used free soluble hApoD monomer purified from cystic fluid. However, in physiological conditions, it is not excluded that ApoD may be able to pass through the BBB via extracellular vesicles (EVs). Indeed, ApoD is found in EVs [24,25,26], including EVs from serum and CSF [24]. Recently, EVs secreted from astrocytes and containing ApoD were shown to be internalized in neurons and to contribute to the survival of neurons under oxidative stress [26]. The passage of EVs from the periphery to the CNS has also been shown, notably in a drug delivery context where EVs were shown to cross the BBB in both directions [74]. Therefore, the high rate at which hApoD was able to exit the brain in vivo may be due to an inclusion on ApoD into EVs by glial cells prior to its passage through the BBB. This could also be a factor explaining the slow rate of internalization observed in vitro in endothelial bEnd.3 cells, where glial cells were absent.

In our study, circulating hApoD did not appear to enter the brain from the circulation. This could indicate that hApoD transport through the CNS barriers is unidirectional or simply follows its concentration gradient (from brain to blood) under normal conditions. It is also possible that the quick pace at which hApoD is internalized by other tissues depletes the circulating stocks of hApoD in a manner that prevents its return to the brain. It is worth mentioning that our data do not definitely prove that ApoD is incapable of entering the brain from the circulation. Our study was aimed at characterizing the biodistribution of hApoD upon release from the brain. Injected radiolabeled proteins that do enter the brain often do so at a low percentage (e.g., approximately 1% of the total injected amount for transferrin) [75]. Therefore, studies centered on the brain often employ methodologies different to that used here. These include the use of a positive control and the injection of proteins into the left jugular vein instead of the tail vein [76], with a possible recirculation of radiolabeled proteins [77]. Further steps will be required to formally ascertain if ApoD can enter the brain from the circulation. ApoD exist in different forms. Much like BSG, ApoD also possess multiple glycosylation levels, which are likely to influence its protein–protein interactions [2]. Glycosylation plays an important role in the thermostability, folding, as well as the overall charge of a protein and can influence its protein–protein interactions [70]. The injection of different forms of ApoD, including monomers, dimers, tetramers, and EV containing ApoD—as well as differently glycosylated ApoD—could also give different results.

In conclusion, our study shows for the first time that hApoD can efficiently exit the brain, most likely by active passage through the BBB and reach peripheral tissues with functions related to lipid and glucose homeostasis (liver, muscles, pancreas, adipose tissues, and intestines, together known as metabolic organs) as well as excretion (Kidneys/urine). The characterization of hApoD’s distribution pattern represents an important advance in the understanding of ApoD function and correlates well with its known functions. The high amount of hApoD found in the liver and urine in our experiments suggests that ApoD may have a role in the excretion and recycling of brain lipids. Our results also suggest that ApoD accumulation in peripheral tissues is likely dependent on BSG and influenced by its glycosylation levels. ApoD’s capacity to leave the brain is likely more complex and might not depend on BSG. However, it remains unclear what receptor other than BSG participates in this process. Furthermore, ApoD’s own glycosylation and incorporation into EVs is also likely to influence ApoD cell internalization and tissue distribution.

## 4. Materials and Methods

### 4.1. Animals

Experiments were carried out on C57BL/6 male mice of 3 to 4 months of age. Animals were housed under standard conditions at constant temperature (20–22 °C) and humidity (50–60%), under a 12 h light/dark cycle with free access to water and food (standard rodent chow; Charles River #5075). Experimental procedures were approved by the Animal Care and Use Committee of Université du Québec à Montréal (protocol #962, reference number 0319-962-0320, 24 April 2019).

### 4.2. Protein Radiolabelling

Human ApoD (hApoD), purified from breast cyst fluid [4] and bovine serum albumin (BSA) were both radiolabeled with iodine-125 in the form of iodine monochloride following the method described by A.S. McFarlane [78]. Unbound Iodine-125 was removed by exclusion chromatography using Bio-Spin 6 gel columns (BioRad #732-6002, Mississauga, ON, Canada). The eluted protein concentration was assessed by Bradford assay [79]. Specific radioactivity ranged from 0.11 to 0.16 μCi/μg of protein. Radiolabeled proteins were visualized after denaturing gel electrophoresis followed by revelation using a Molecular Dynamics Storage Phosphor Screen (Kodak Storage Phosphor Screen SO230, Rochester, NY, USA) and Typhoon FLA9500 (Appendix A).

### 4.3. Intracerebroventricular and Intravascular Injections

For intracerebroventricular (ICV) injection, ketamine/xylazine anesthetized animals were placed on a stereotaxic table (Stoelting #51600). Approximately 2.5 μg of radiolabeled hApoD or BSA (5 μL volume; 7 × 10^5^ CPM) was bilaterally injected into the lateral ventricles at a rate of 0.2 μL/min (Hamilton syringe 1701 N) as previously described [80,81]. The bregma coordinates used for injection were: 1.0 mm lateral, −0.3 mm posterior and −2.5 mm below, as previously described [82]. The needle was gently removed 5 min after the end of each injection. For intravascular injections, 100 μL PBS (pH 7.4) containing 3 × 10^6^ CPM (1.35 μCi) of either radiolabeled hApoD or BSA was administered via the tail vein (26G syringe).

### 4.4. Tissue and Fluid Sample Preparation

To collect urine, the bladder was emptied with a 26G needle and syringe. Blood was collected by cardiac exsanguination from the left ventricle with a 22G needle and syringe without damaging the inferior vena cava. Immediately after, the blood was flushed from the circulation by performing a whole-body perfusion. Briefly, a catheter was inserted into the inferior vena cava from the right auricle of the heart and maintained in place with chirurgical sutures. The lower part of the right ventricle was perforated to allow the blood to escape. A buffer solution (128 mM NaCl, 4.0 mM KCl, 0.62 mM H_2_PO_4_, 1.1 mM NaH_2_PO_4_, 11.1 mM Dextrose, 10.1 mM HEPES, 1.1 mM MgCl_2_, 0.42 mM MgSO_4_, 1.5 mM CaCl_2_, pH 7.4, 37 °C) was then circulated with a peristaltic pump at 2 mL/min for at least 5 min, until the buffer-diluted blood coming out was clear. Organs and tissues were then collected: brain, liver, adductor and medial hamstring muscles, kidneys, heart, spleen, pancreas, caecum, bone marrow, as well as omental, mesenteric, and subcutaneous adipose tissues.

Plasma was prepared by centrifugation (15 min; 2000× *g*) of 500 μL blood mixed with 50 μL EDTA 10%. Whole organ weight was determined whenever possible (for brain, liver, kidneys, heart, spleen, pancreas, and caecum). Because only samples of adipose tissues and bone marrow were obtained, their whole organ weight was not determined. Tissue samples were homogenized in a lysis buffer (2 μL/mg of tissue) suitable for tissues with high a lipid content (50 mM Tris-HCl pH 7.4, sucrose 250 mM, 100 mM NaF, 10 mM sodium pyrophosphate, 1 mM EDTA, 1 mM DTT, 1 mM sodium vanadate, 1 mM PMSF) using a Dounce tissue grinder (Wheaton, 357421). Tissue homogenate and fluid (plasma and urine) radioactivity was measured by scintillation counting. Individual organ radioactivity was extrapolated by multiplying the specific radioactivity (CPM/g of tissue sample) by the weight of the organ. To extrapolate blood radioactivity, plasma was considered to account for 58% of the blood volume (Feher, 2012) and the total blood volume was considered to be 2 mL for a 25 g mouse [83]. Pelleted blood cells had radioactive levels approximately half of that of the plasma itself according to Geiger counter readings. Therefore, blood cell specific radioactivity (*BCSR*) was estimated from the plasma specific radioactivity according to the following equation:$$BCSR=\frac{Plasma\;specific\;radioactivity}{2}$$
Total blood radioactivity (*TBR*) was calculated as follows:$$TBR=\left(\left(Plasma\;specific\;radioactivity\times 0.58\right)+\left(BCSR\times 0.42\right)\right)\times \frac{2\;\mathrm{mL}}{25\;g}\times Body\;weight$$

Similarly, whole muscle radioactivity was extrapolated from the combined adductor and medial hamstring muscles specific radioactivity and the animal body weight. Total wet skeletal muscle mass was considered to be 6.9 g for a 25 g mice, consisting of 1.8 g of dry weight [84] and 5.1 g of water (water composition of 75%) [85]. Total muscle radioactivity was calculated as follows:$$Total\;muscle\;radioactivity=Muscle\;radioactivity\times \frac{6.9\;g}{25\;g}\times Body\;weight$$

In addition to measuring radioactivity from the urine contained in the bladder, the bottom of each cage was swabbed with a humid absorbing paper and this radioactivity (measured by scintillation counting) was added to the urine fraction. The total recovered radioactivity was determined by adding all tissues and fluids together.

### 4.5. Cell Culture

The bEnd.3 brain endothelial cell line was obtained from the American Type Culture Collection (ATCC, CRL-2299, Manassas, VA, USA). Cells were cultured in Dulbecco’s modified Eagle’s medium (DMEM), with 10% fetal bovine serum (FBS), 1% sodium pyruvate (1 mM), and antibiotics (100 U/mL penicillin and 100 µg/mL streptomycin) from Wisent Bioproducts, in a humidified incubator at 37 °C with 5% CO_2_.

To generate monolayers, 30,000 bEnd.3 cells were seeded on the upper surface of Falcon cell culture inserts (Corning, #08770, Corning, NY, USA) placed within the wells of a 24-well plate (Corning, #09761146). The culture medium (top and bottom compartments) was replaced every two days. After 7–10 days, monolayer imperviousness was systematically validated via transendothelial electrical resistance (*TEER*) (Appendix A) measurements using a voltage and resistance meter (EVOM2, World Precision Instruments Inc., Sarasota, FL, USA) equipped with a cell culture cup chamber electrode (Endohm-6, World Precision Instruments Inc. Sarasota, FL, USA). The background value measured on a cell-free insert (insert resistance) was subtracted from each raw *TEER* measurement. Resistivity (Ohms·cm^2^) was calculated as follows:$$TEER=\left(Resistance\;\left(\Omega \right)-Insert\;Resistance\;\left(\Omega \right)\right)\times Surface\;area\;\left(cm^{2}\right)$$

Monolayers with *TEER* values of at least 35 Ohms/cm^2^ (Appendix A) were retained for transcytosis experiments (Wuest, Wing, and Lee 2013; Clark and Davis, 2015).

The monolayers were also systematically validated by assessing permeability to Dextran-FITC (10 kD, Sigma-Aldrich, St-Louis, MO, USA). Briefly, 10 µL of Dextran-FITC (1 mg/mL; in serum-free, phenol red-free culture media) were added to the media on the top of the insert. After a 1 h incubation, 150 µL of culture medium was sampled from the bottom compartment and analyzed on a 96-well plate reader (Tecan, Switzerland). Apparent permeability index (Papp) was calculated from the following equation:$$P_{app}=\frac{V_{R}\Delta C_{R}}{\Delta t\;S_{ins}C_{D}}$$

With Papp apparent permeability (cm·s^−1^); *V_R_*, volume of receiving compartment (cm^3^), Δ*C_R_*, change in concentration in the receiving compartment (µM); Δ*t*, time in seconds (s); *S_ins_*: surface of the insert (cm^2^) and *C_D_*: concentration in the donor compartment (µM) [38]. Monolayers with Dextran-FITC apparent permeability values (Papp) below 10–6 cm·s^−1^ were used for transcytosis experiments [86]. Monolayers were further validated following transcytosis experiments by verifying the presence of Claudin 5-positive tight junctions between cells [38].

For our transwell assays, monolayer cells were treated 24 h on the top side with hApoD (purified from human cystic fluid) at 200 or 400 ng/mL with or without recombinant human cyclophilin A (R&D systems, 3589-CAB). Cell viability (Appendix A) was confirmed using a resazurin-based kit (Sigma-Aldrich, TOX8, St-Louis, MO, USA).

### 4.6. Immunofluorescence

Monolayer cells were fixed with 4% paraformaldehyde 15 min at room temperature, permeabilized with 0.2% Triton X-100 in PBS and blocked with 5% no-fat dry milk in PBS. Primary antibodies rabbit anti-Claudin 5 (Thermofisher #34-1600, Waltham, MA, USA) and mouse anti-ApoD (2b9 1:100) [87] were used with the dilution (1:100). Secondary antibodies rabbit Alexa Fluor 488 (Invitrogen #A11008, Carlsbad, CA, USA) and mouse Alexa Fluor 594 (Invitrogen #A11005) were used with the dilution 1:500. The nuclei were stained with Hoechst 33258 (10 μg/mL, Sigma-Aldrich #861405) 15 min at room temperature. Then, inserts were placed on a slide with mounting medium (Prolong™ Gold Antifade Mountant). Mounted inserts were examined on a Zeiss LSM780 system confocal microscope equipped with a 30-mW, 405-nm diode laser, a 25-mW 458/488/514 argon multiline laser, a 20-mW DPSS 561-nm laser, and a 5-mW HeNe 633-nm laser mounted on Zeiss Axio Observer Z1, as well as a Plan-Apochromat 63× oil DIC 1.4NA objective. Images (1.6× zoom scans) were acquired with Zen 2011 software (Zeiss, Germany). Images were analyzed with ImageJ software (version 1.52p, National Institutes of Health, Bethesda, MD, USA).

### 4.7. Immunoblotting

For tissue sample analysis, lysates were incubated 30 min at 4 °C, cleared by centrifugation (10,000× *g*, 15 min) and the lipid layer was discarded. For bEnd.3 transwell media analysis, media proteins from top and bottom compartment were concentrated with Amicon Ultra-0.5 10 kDa centrifugal filters (Millipore Sigma #UFC501024, Burlington, MA, USA).

For both sample types, protein concentration was assessed by Bradford assay [79]. Proteins (20 μg, unless otherwise indicated) were separated on 10% SDS-PAGE gels and transferred on PVDF membranes. Blocking was performed using 5% milk, 1 h at room temperature. Membranes were then incubated with primary antibodies overnight at 4 °C. Antibodies used were against Basigin (Abcam, ab188190 1:5000), ApoD (MyBioSource #2003092 1:1000), and goat anti-rabbit HRP conjugated IgG (Sigma-Aldrich #A6154 1:10,000 or Abcam ab6721 1:4000). Bands were visualized using chemiluminescent HRP substrate (Millipore, WBKLS0500, Burlington, MA, USA) in a FUSION FX7 Imaging system and analyzed with the gel analyzer function of Image J software (National Institutes of Health, Bethesda, MD, USA).

### 4.8. Statistics

Our histogram results are presented as mean ± standard error of the mean. The statistical analysis were performed with the GraphPad 7 software. For histograms, statistically significant differences from control values were determined by using a one-tailed Student’s *t*-test and a Welch’s correction was applied when variances between groups were unequal. Variance equivalence between groups was determined by Fisher’s f-test. Associations between variables of interest were quantified by linear regression using Pearson’s product moment correlation coefficients analysis. Statistical significance was considered reached if the *p*-value was <0.05.

## Figures and Tables

**Figure 1 ijms-22-04118-f001:**
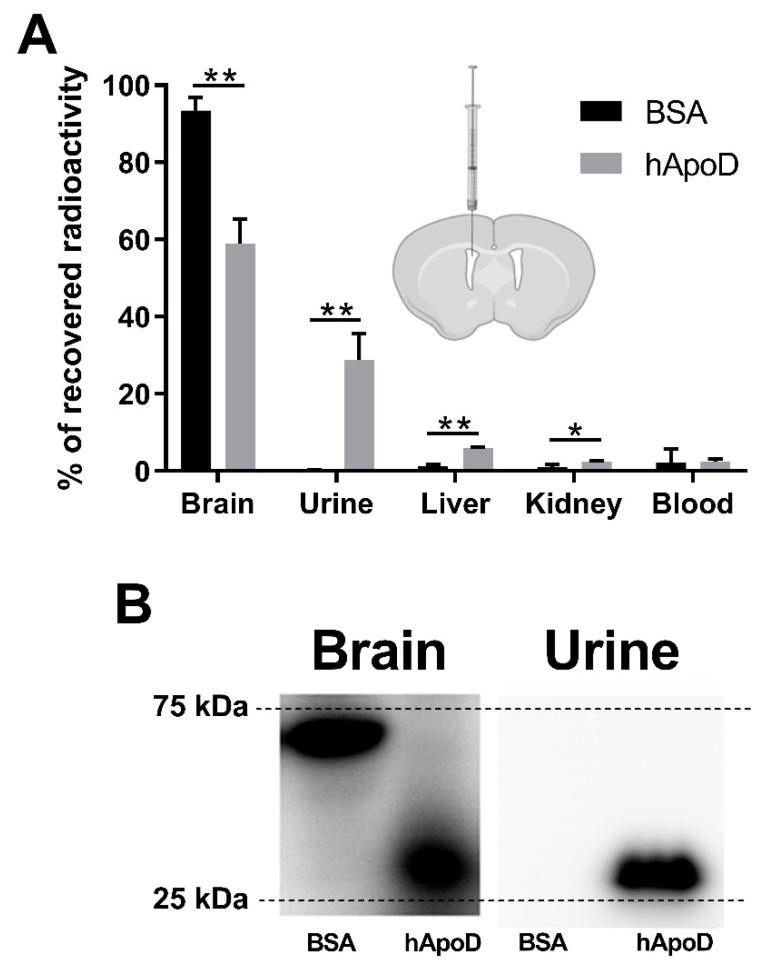
Human Apolipoprotein D (hApoD) injected in the brain accumulates in urine and liver. Radiolabeled proteins (Bovine Serum Albumin (BSA) and hApoD) were injected in mouse cerebral ventricles. Blood, urine, and—after perfusion—organs were collected 3 h post injection. (**A**) Radioactivity recovered in fluids and tissue homogenates (%) compared to the total radioactivity recovered in the whole body. Statistical significance was evaluated via a Student *t*-test: * *p* < 0.05, ** *p* < 0.01 (n = 3 animals). (**B**) Radiolabeled proteins extracted from brain and urine and visualized by radiography. A representative electrophoresis gel is shown.

**Figure 2 ijms-22-04118-f002:**
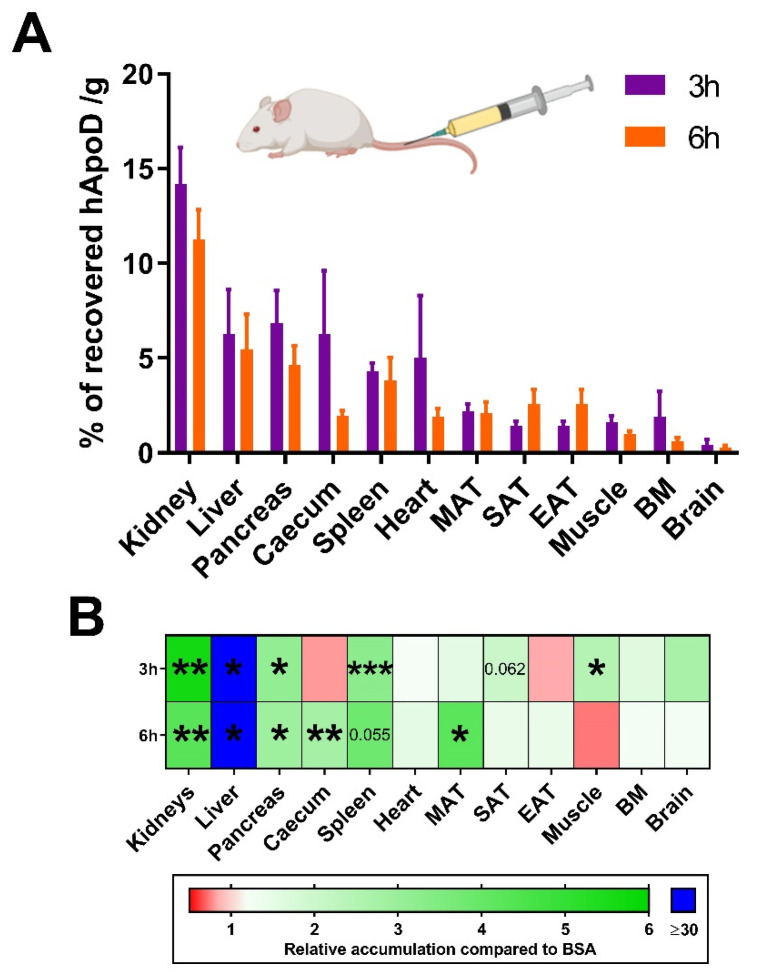
Circulating hApoD accumulates in specific peripheral organs. Radiolabeled proteins (Bovine Serum Albumin (BSA) and hApoD) were injected intravenously in mice. Blood, urine, and—after perfusion—organs were collected 3 and 6 h post-injection. (**A**) Results are presented as the percentage of radioactivity recovered per gram of tissues. EAT: epididymal adipose tissue, SAT: subcutaneous adipose tissue, MAT: mesenteric adipose tissue, and BM: bone marrow. (**B**) The heat map shows the accumulation of hApoD relative to albumin. Statistical significance was evaluated via a Student *t*-test: * *p* < 0.05, ** *p* < 0.01, *** *p* < 0.001 (n = 4 animals).

**Figure 3 ijms-22-04118-f003:**
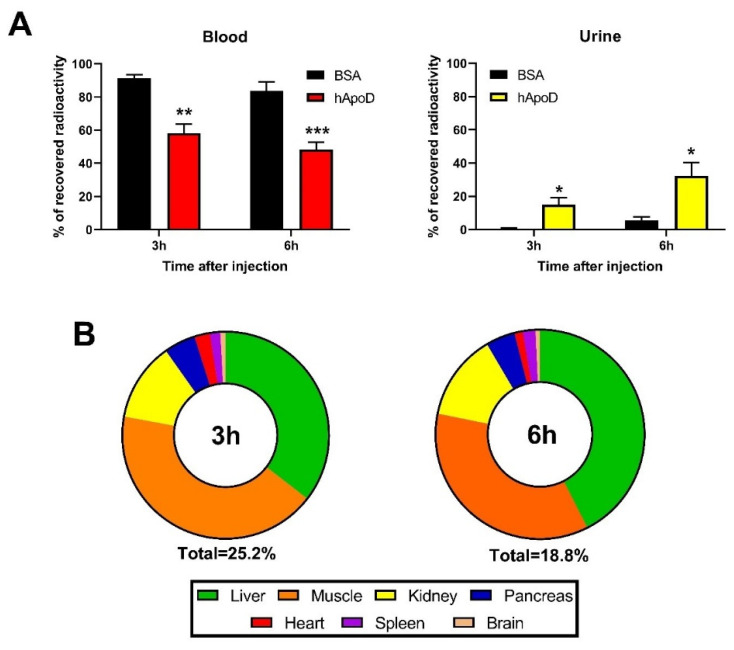
Relative accumulation of hApoD in fluids and tissues. Radiolabeled proteins (Bovine Serum Albumin (BSA) and hApoD) were injected intravenously in mice. Blood, urine, and—after perfusion—organs were collected 3 and 6 h post-injection. Results are presented for each (**A**) fluid and (**B**) organs as their average respective percentage of the total radioactivity recovered in the animals. Statistical significance was evaluated via a Student t-test: * *p* < 0.05, ** *p* < 0.01, *** *p* < 0.001 (n = 4 animals).

**Figure 4 ijms-22-04118-f004:**
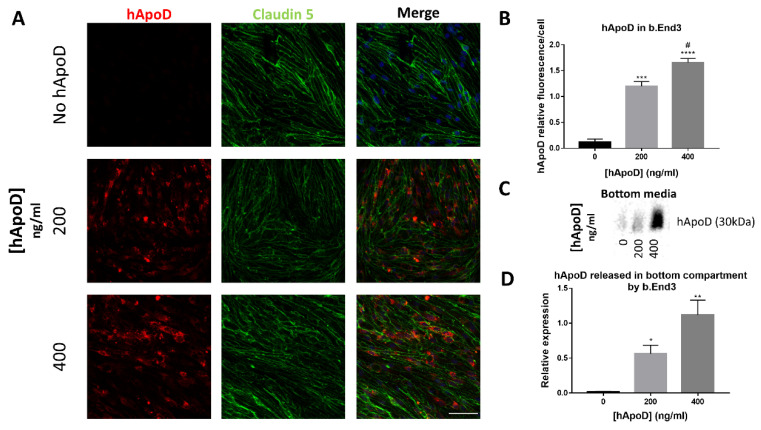
hApoD is subject to endocytosis and transcytosis through bEnd.3 cells. The top side of bEnd.3 cell monolayers was exposed to different concentrations of hApoD for 24 h. (**A**) Cells were immunostained for hApoD (red) and a tight junction marker (Claudin-5, green). Nuclei were stained with Hoechst (blue). Scale bar: 50 μm. (**B**) Bottom media proteins were immunoblotted for hApoD. Representative images are shown (n = 3 independent experiments). Statistical significance for panels (**B**,**D**) was evaluated via a Student *t*-test: * *p* < 0.05, ** *p* < 0.01, *** *p* < 0.001, **** *p* < 0.0001 against the 0 ng/mL (*) and 200 ng/mL (#) conditions (**A**–**D**).

**Figure 5 ijms-22-04118-f005:**
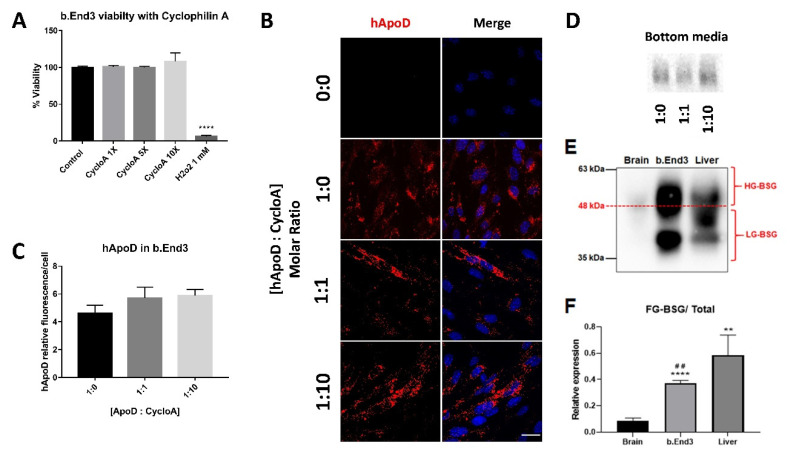
Cyclophilin A competition does not reduce hApoD internalization in bEnd.3 cells. (**A**) Viability of bEnd.3 cells in the presence of cyclophilin A, at 1, 5, and 10 times the molar concentration of hApoD (13.33 nM) used in the experiment. H_2_O_2_ was used as a control for the loss of cell viability. (**B**) The top side of bEnd.3 cell monolayers were exposed to hApoD (13.33 nM) for 24 h. Cyclophilin A was included from the beginning of the experiment at a molar equivalent (1:1) or in molar excess (1:10) relative to hApoD. Cells were immunostained for hApoD (red). Nuclei were stained with Hoechst (blue). Scale bar: 20 μm. Representative confocal sections are presented. (**C**) Average hApoD signal per cell (5 images taken for each monolayer). Fluorescence was normalized by the number of cell nuclei. (**D**) Bottom media proteins immunoblotted for hApoD. (**E**) Immunoblot of BSG in bEnd.3 and control tissues (HG-BSG, Highly glycosylated BSG; LG, lowly glycosylated BSG). Panels (**D**,**E**) are representative results. (**F**) Quantification of LG-BSG expression relative to total BSG expression as compared to the brain (*) and liver (#). Statistical significance for panels (**A**,**C**,**F**) was evaluated via a Student *t*-test: ** and ^##^
*p* < 0.01, **** *p* < 0.0001 (n = 3 independent experiments).

**Figure 6 ijms-22-04118-f006:**
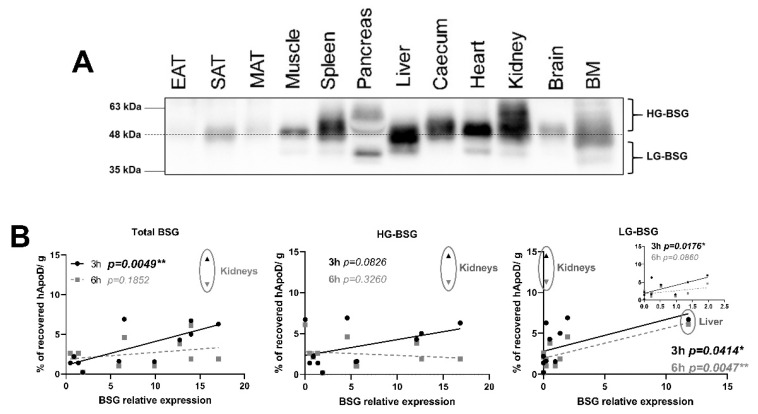
hApoD tends to accumulate in tissues expressing underglycosylated Basigin. (**A**) Immunoblot of Basigin (BSG) in its multiple glycosylated forms (HG, highly glycosylated; LG, lowly glycosylated) in various tissues, including epididymal adipose tissue (EAT), subcutaneous adipose tissue (SAT), mesenteric adipose tissue (MAT), and bone marrow (BM). Each well contained whole protein extracts from 20 μg of tissue. A representative immunoblot is shown (n = 4 animals). (**B**) Correlations between average hApoD accumulation (% of recovered radioactivity per gram of tissue) and average expression level of different classes of glycosylated Basigin in various tissues, 3 (black) and 6 h (grey) after IV injection. Inset: liver data eliminated from the correlation. Pearson’s product moment correlations are presented: * *p* < 0.05, ** *p* < 0.01 (n = 4 animals).

## Data Availability

Not applicable.

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
