# Peer review of "Cerebral Apolipoprotein D Exits the Brain and Accumulates in Peripheral Tissues"

_ijms, 2021, doi:10.3390/ijms22084118_

Round 1

Reviewer 1 Report

Manuscript Number: ijms-1170873

Cerebral Apolipoprotein D exits the brain and accumulates in 2 metabolic tissues by Frederik Desmarais et al.

The paper has described that apolipoprotein D (ApoD) can quickly and efficiently exit the CNS and reach the liver and kidneys/urine, organs linked to the recycling and excretion of lipids and toxins by using radiolabeled human ApoD.

This present result is interesting, and some specific point should be considered.

Major comments

The authors should clarify the findings that ApoD, as described in the discussion section, is involved in the removal of waste products such as oxidised lipids and inflammatory substances from CNS.

The authors describe that apoD is involved in the removal of waste products from the CNS, while accumulating and acting on not only in the liver and kidneys, but also in other organs such as muscle, intestine and spleen. These logics seem to be inconsistent.

Are there any results or findings that ApoD is present on lipoprotein in CNS, circulating blood, or peripheral tissues?

Reviewer 2 Report

The manuscript entitled "Cerebral Apolipoprotein D exits the brain and accumulates in 2 metabolic tissues" is well written and the primary scientific intention is interesting. While ApoD is produced in the brain, it has several demonstrated effects in peripheral tissues.
The authors designed their study to determine if cerebral ApoD can reach the circulation and accumulate inperipheral tissues.
To demonstrate that ApoD can exit the brain cerebral ventricles of mice were injected with radiolabelled hApoD. The results for 3 h are presented in fig. 1 but no comment is done for a 6h diffusion. In addition a complete radiography should be presented for the urine sample the spot is cutted.
The distribution in different tissues was estimated by intravenous injection that simplifies the study. The authors retained two data points at 3h and 6h and determined that, once in the blood stream, ApoD accumulates mainly in liver and kidneys. Do the authors have information about a 24h data point?  The reviewer is surprised by the use of the term "metabolic organs". This term was used in number of published articles, however it may be confusing: do the authors means "energy metabolism related organs/tissues ?
Another concern is about the use and interpretation of the data with the bEnd.3 cells. The authors stated that the cells are not polarized, but consider that they performed their study from the apical to basal side... The impermeability of the model was established by electrical resistance measurement and dextran permeability and by expression of Claudin 5. Claudin 5 is known in polarized cells to be localized at the apical side. The use of confocal microscopy could clarify this aspect.  In addition the immunostaining presented fig 4 did not attest the integrity of the monolayer.
The authors have finally tried to demonstrate that cyclophilin A competition does not reduce hApoD internalization in bEnd.3 cells. This part is very confusing since the abbreviations used were not defined (LG-BSG, HG-BSG, LG-glycosylated forms, underglycosylated BSG expression, lowly-glycosylated BSG, high LG-BSG...). This should be re-tidy-up!
Additional data must be provided and inconsistencies must be resolved for the manuscript to become acceptable for publication.

Round 2

Reviewer 1 Report

Authors should show how much of the apoD is free presence and how much is bound to lipoproteins in both CNS and circulating blood in this paper.

Reviewer 2 Report

The reviewer thanks the authors for the quality of the revised manuscript. All the comments/questions raised by the reviewer were adequately addressed. From their point of view, the manuscript is now acceptable for publication.